# The Nutrition-Microbiota-Physical Activity Triad: An Inspiring New Concept for Health and Sports Performance

**DOI:** 10.3390/nu14050924

**Published:** 2022-02-22

**Authors:** Nathalie Boisseau, Nicolas Barnich, Christelle Koechlin-Ramonatxo

**Affiliations:** 1Laboratoire des Adaptations Métaboliques à l’Exercice en Conditions Physiologiques et Pathologiques (AME2P), Université Clermont Auvergne, CRNH Auvergne, 63000 Clermont-Ferrand, France; 2M2iSH, UMR 1071 Inserm, USC-INRAE 2018, Microbes, Intestin, Inflammation et Susceptibilité de l’Hôte (M2iSH), Université Clermont Auvergne, CRNH Auvergne, 63000 Clermont-Ferrand, France; nicolas.barnich@uca.fr; 3DMEM, Université de Montpellier, INRAE, 34000 Montpellier, France; christelle.ramonatxo@umontpellier.fr

**Keywords:** exercise, training, dysbiosis, eubiosis, diets, supplements, competitive microbiota, inter-organ crosstalk, sedentary, athletes, athletic performance, health

## Abstract

The human gut microbiota is currently the focus of converging interest in many diseases and sports performance. This review presents gut microbiota as a real “orchestra conductor” in the host’s physio(patho)logy due to its implications in many aspects of health and disease. Reciprocally, gut microbiota composition and activity are influenced by many different factors, such as diet and physical activity. Literature data have shown that macro- and micro-nutrients influence gut microbiota composition. Cumulative data indicate that gut bacteria are sensitive to modulation by physical activity, as shown by studies using training and hypoactivity models. Sports performance studies have also presented interesting and promising results. Therefore, gut microbiota could be considered a “pivotal” organ for health and sports performance, leading to a new concept: the nutrition-microbiota-physical activity triad. The next challenge for the scientific and medical communities is to test this concept in clinical studies. The long-term aim is to find the best combination of the three elements of this triad to optimize treatments, delay disease onset, or enhance sports performance. The many possibilities offered by biotic supplementation and training modalities open different avenues for future research.

## 1. Introduction

It has been suspected that gut microbiota may have a role in health and sports performance for a very long time. The massive sequencing of gut microbiota specimens in the 2010s, thanks to the technological advancements in high-throughput sequencing and bioinformatics analyses, and more recently, the development of methods to quantify different microbial metabolites, allowed population-level studies to be carried out on the human microbiota. Their findings help to better understand the microbiota’s role in physiology, its functional imbalance in various chronic pathologies [1], and its implications in athletic performance. In addition, mechanistic studies with gnotobiotic model organisms have brought novel insights into the underlying molecular mechanisms and can be used to test strategies to modulate the gut microbiota composition. In this review, we first summarize the knowledge on the mechanisms underlying the implications of gut microbiota in various chronic pathologies and sports performance. Then, we focus on the combination of physical activity and nutritional interventions to modulate gut microbiota composition in the context of health and performance. Our hypothesis is that the interaction of their underlying mechanisms might potentiate their effects. This new knowledge could be used to develop strategies (i.e., diet changes, supplementation, physical activity programs) to modulate the gut microbiota with the ultimate aim of preventing and/or treating various pathologies or improving performance in elite athletes.

## 2. Gut Microbiota: What Is It?

The term intestinal (or gut) microbiota describes the different microorganisms that live in the digestive tract. Natural microbiotas can also be found in other areas of the body, such as the skin, urogenital system and respiratory system; however, the gut microbiota is the densest and most studied. Since its first description by Antonie van Leeuwenhoek, who, in 1681, observed under the microscope more than 1000 “animalcules” in his feces, evidence about its essential role in human health has been accumulating [2].

The gut microbiota is mainly composed of bacteria (10^13^ bacteria), but eukaryotes (fungi and protozoa) and archaea are found occasionally. Similarly, viruses that infect bacteria (called “phages”) are frequently detected in the microbiota. They can modify the abundance, genetic profile and gene expression of bacterial populations. Thus, the “virome” is undoubtedly another piece in the puzzle of gut microbiota physiopathology. The characterization of the genome of all microorganisms found in the intestine (the intestinal metagenome) by high-throughput sequencing identified thousands of different species, mostly bacteria, and over 3 million genes, which correspond to 150 times the human genome [1,3,4,5]. The gut microbiota can reach up to 1.5 kg in a 70 kg individual [6]. Like the fingerprints, the intestinal microbiota is unique to each individual in terms of quality and quantity. Among the 160 species of bacteria that correspond to the average microbiota of a healthy individual, only half are commonly found in different individuals, and only 15 to 20 species are present in all human beings. These species are in charge of the essential functions of the microbiota. Indeed, although the microbiome composition varies among individuals, metabolic functions are incredibly stable and conserved [7].

The gut microbiota density varies along the gastrointestinal (GI) tract. It is fairly low in the stomach, duodenum and jejunum and increases in the ileum and colon. The stomach is characterized by the presence of oxygen and high acidity. It selectively hosts acid-tolerant and facultative anaerobic microorganisms, such as lactobacilli, streptococci and yeast species. In the small intestine, the microbiota is mainly composed of facultative anaerobic bacteria (e.g., *Lactobacilli, Streptococci* and *Enterobacteria*) and strict anaerobic bacteria (*Bifidobacteria, Bacteroides* and *Clostridia*). In the colon, the slower intestinal peristalsis and anaerobiosis favor the encroachment of a complex microbiota with the highest bacterial diversity (more than 1000 bacterial species) and density [8,9]. Two bacterial phyla account for 80–90% of the bacterial population in the colon: Firmicutes and Bacteroidetes [10,11,12,13]. Proteobacteria and Actinobacteria are minority phyla. Overall, in human adults, the dominant bacterial genera are *Bacteroides*, *Eubacterium*, *Ruminococcus*, *Clostridium,* and *Bifidobacterium*. The intestinal microbiota composition also varies according to the species and within the same GI segment between the near-mucosal compartment and the digesta/stool. For instance, in the colon, Firmicutes of the *Lachnospiraceae* and *Ruminococcaceae* families predominate in the near-mucosal compartment, whereas Bacteroidetes of the *Bacteroidaceae*, *Prevotellaceae*, and *Rikenellaceae* families are preponderant in the digesta/stool [14]. Murine models are widely used in basic/preclinical research, but the taxonomic differences between human and mouse microbiota do not always allow the transfer of the acquired data to the clinic [15].

The gut microbiota, often described as the “forgotten organ,” has many beneficial functions in the organism when it is in symbiosis with the host [6]. The microorganisms in the microbiota play a direct role in digestion, for instance, by ensuring the fermentation of substrates and non-digestible food residues, by facilitating the nutrient assimilation thanks to a set of enzymes that are not present in human cells, and by participating in the synthesis of some vitamins. They also influence the overall functioning of the GI tract and participate in the functioning of the intestinal immune system, which is essential for the intestinal wall barrier function. The host-microbiota symbiotic interactions reflect their co-evolution with reciprocal benefits. In conclusion, the human gut microbiota is a complex ecosystem that is different from that of other microbiota types in the human body [7]. It has adapted to local environmental constraints through mechanisms of co-evolution between microorganisms and hosts. Due to the complexity of this ecosystem and of its functions, host-microbiota interactions represent a fragile equilibrium that can be disrupted in many pathologies.

## 3. Gut Microbiota in Health and Disease

Locally, the gut microbiota is involved in the intestinal homeostasis by participating, for example, in the barrier function, extraction of nutrients from the diet, and conjugation of bile acids. However, its contribution is not limited to the intestine but also concerns, among others, the metabolism, immune system and immune responses [16,17]. The intestinal microbiota contribution to our health is becoming increasingly well known. Besides its homeostatic role, changes in the microbiota composition, called dysbiosis, are implicated in the establishment and chronicity of various pathologies, such as chronic inflammatory bowel disease (IBD), irritable bowel syndrome, colorectal cancer, metabolic diseases (type 2 diabetes, obesity), depression, and cardiovascular diseases [18] (Figure 1). Dysbiosis, i.e., the qualitative and/or functional alteration of the intestinal microbiota, can contribute to the etiology of some diseases, especially those in which autoimmune or inflammatory mechanisms are implicated.

The intestinal microbiota plays a role in the natural inflammation observed in the digestive tract. Inflammation is an important biological process closely related to immunity. A physiological low inflammation level is essential for immune activation and allows the microbiota to be controlled. Conversely, important inflammatory reactions are triggered by the presence of pathogenic species [19,20] through a mechanism based on the presence of inflammatory bacterial components, such as lipopolysaccharides (LPS) on the surface of Gram-negative bacteria. These antigens induce an immune response that leads to the production of pro-inflammatory mediators (cytokines) by macrophages in the intestine. Local inflammation is triggered, and the intestinal wall permeability increases. LPS can then pass through the intestinal wall, enter the bloodstream, and cause systemic inflammation in other target tissues [21].

### 3.1. A Clear Link with GI Diseases

Chronic IBDs, such as Crohn’s disease and ulcerative colitis, are the consequence of the inappropriate activation of the immune system in the intestine. The intestinal microbiota may be implicated because symptoms improve in patients undergoing antibiotic treatment and because inflammatory intestinal lesions disappear in people whose intestinal wall is no longer in contact with the feces (following the installation of fecal diversion) [22]. This is probably due to the role of intestinal bacteria and their metabolites in the local immune response balance [23]. In terms of microbiota composition, the abundance of bacteria belonging to the phylum Firmicutes is decreased, whereas that of bacteria belonging to the phylum Proteobacteria is increased [24]. *Faecalibacterium prausnitzii*, one of the dominant bacteria in the Firmicutes phylum and in the human intestinal microbiota, appears to be particularly decreased in patients with IBD, especially Crohn’s disease, and its level is predictive of the risk of relapse [25,26]. Furthermore, in patients with IBD, dysbiosis may favor the colonization by and growth of potentially pathogenic and pro-inflammatory microorganisms. Several microorganisms have been implicated in Crohn’s disease: *Candida albicans*, *Listeria monocytogenes*, *Mycobacterium avium* subspecies paratuberculosis, enterotoxin-producing *Bacteroides fragilis*, and adherent and invasive *Escherichia coli* [27,28,29].

### 3.2. Cancer

Normally, there is a balance between microbiota quality, immune system efficiency, and intestinal barrier integrity. Dysbiosis can promote cancer onset and/or progression through different mechanisms:(i) The presence of specific microorganisms or of intestinal dysbiosis has been associated with some tumors. First, the pathogen can cause DNA lesions, for example, through the production of genotoxins, as reported for *Helicobacter pylori*, a bacterium that increases the risk of gastric cancer [30,31], and colibactin-producing *E. coli*, a bacterium that increases the risk of colorectal cancer [32,33,34]. Second, the microbiota imbalance favors some species (*Fusobacterium*) that can abnormally stimulate oncogenic pathways, such as the beta-catenin signaling pathway [35];(ii) The close interaction between microbiota and local immunity [36,37]. Several pro-inflammatory or immunosuppressive signaling pathways are activated in the presence of dysbiosis. Moreover, dysbiosis increases intestinal permeability, allowing the passage of oncogenic compounds from the intestinal lumen into the body;(iii) Microbiota anomalies might lead to the induction of genes linked to cancer cell survival [32], thus promoting tumor progression;(iv) More recently, bacteria have been identified within tumors. Understanding their nature, origin, and influence on cancer development/progression may provide new therapeutic avenues [38]

### 3.3. Metabolic and Cardiovascular Diseases

Cardiovascular and cerebrovascular diseases (e.g., atherosclerosis, hypertension, stroke) and metabolic diseases (diabetes, obesity) have multifactorial (genetic, nutritional and environmental) origins. The role of each of these factors varies from one individual to another, and the implicated molecular mechanisms remain to be precisely described. However, it is becoming increasingly clear that the intestinal microbiota plays a role in their genesis [39]. For instance, upon fecal transplantation of microbiota from obese mice, axenic (i.e., germ-free) mice significantly and rapidly gain weight [40]. Several mechanisms could be at the origin of these relationships. In diabetes and obesity, chronic inflammation is favored by the higher percentage of fat in the diet that increases the proportion of Gram-negative bacteria in the intestine, thus increasing the local level of inflammatory LPS. LPS can pass in the bloodstream and reach other organs/tissues (e.g., liver, fat, muscle), where they favor the installation of low chronic inflammation that will promote the development of insulin resistance, a prerequisite for diabetes and obesity [41].

In addition, some bacterial metabolites might have a determining role in the development of cardiometabolic diseases, such as type 2 diabetes, atherosclerosis and arterial hypertension. The most convincing findings concern trimethylamine. This waste product produced by the microbiota can pass into the bloodstream. It is then oxidized by the liver to trimethylamine-N-oxide, a substance that promotes the formation of atherosclerotic plaques [42,43].

### 3.4. Brain

The nervous system that regulates the intestine (enteric nervous system, ENS) contains about 200 million neurons. Its primary function is to ensure intestinal motricity. Furthermore, the intestine is in close and bidirectional interaction with the central nervous system (CNS). This explains why the ENS is referred to as the second brain. A microbiota imbalance could modify the information transmitted to the CNS and ENS, thus altering the functioning of both organs [44]. Multiple mechanisms may be involved. Compounds from the microbiota (metabolites or structural elements) can diffuse through the intestinal wall and directly modulate the ENS. This affects the functioning of the intestine, the vagus nerve, and, indirectly, the brain. These compounds can also directly reach the CNS via the bloodstream. In the brain, they can negatively affect some functions directly or after metabolization. Finally, bacteria can indirectly modulate some endocrine functions that are controlled by the CNS by interacting with enteroendocrine cells located in the gut and linked to the brain, as described for the serotonin pathway [45,46]. These data support the hypothesis that the intestinal dysbiosis observed in neurodevelopmental disorders and neurodegenerative diseases, such as Parkinson’s and Alzheimer’s disease, may contribute not only to the digestive disorders described in these patients but also to their neurological symptoms.

### 3.5. Towards Individualized Treatments

Trials in patients with obesity, metabolic syndrome, or Crohn’s disease have shown that fecal transplantation improves some biological parameters, but the effect remains very modest [47,48]. Therefore, it has been proposed that research should focus on developing personalized interventions adapted to each patient’s condition and microbiota characteristics. For instance, as microbiota may synergize with some drugs, particularly anti-cancer compounds, these drugs could be combined with interventions to restore the proper microbiota functioning (e.g., probiotics, fecal transplantation, metabolites) [49,50]. In the near future, predictive tests of the patients’ response to a given treatment may be developed based on their microbiota analysis. In this context, it is essential to study the factors that influence the human gut microbiota composition, particularly diet and physical activity.

Take home messages, as well challenges and future directions dealing with the human microbiota in health and disease are summarized in the Box 1.

Box 1The Human Microbiota in Health and Disease.
✓The gut microbiota is a real “orchestra conductor” in the host’s physio(patho)logy.✓Dysbiosis is observed and is implicated in many chronic diseases.✓Microbiota may synergize with some drugs and modulate their efficacy in chronic diseases.✓Future microbiota-based tests to predict each patient’s response to a drug.✓Studies on diet and physical activity, as gut microbiota composition regulators, are needed.


## 4. Diet Influences the Gut Microbiota Composition

The intestinal microbiota and the associated metabolic products interact with the host in many different ways, influencing gut homoeostasis and health outcomes. Studies in mice and humans suggest that the modern Western lifestyle, particularly the high-fat and/or high-sugar diets, can persistently alter commensal microbial communities, leading to microbial disturbances. This favors pathogen susceptibility [51], obesity [40,52], auto-inflammatory diseases [53], and other pathologies. Diet might explain more than 50% of the microbial structural changes in mice and 20% in humans, highlighting the potential of dietary strategies in the management of metabolic diseases through gut microbiota modulation [54,55]. Normally, the human microbiota remains stable for months and possibly for years [56]. In humans, short-term dietary interventions can rapidly modify microbiota diversity, but these changes are only transient [57]. 

The amount, type, and balance of the main dietary macronutrients (fat, proteins, and carbohydrates) greatly influence the intestinal microbiota. 

### 4.1. Fats

Consumption of a high-fat diet (HFD) significantly reduces the fecal concentration of short-chain fatty acids (SCFA), including butyrate, and of *Bifidobacteria*, compared with a low-fat diet (Brinkworth et al., 2009). Moreover, several human studies demonstrated that HFDs increase the total anaerobic microflora and *Bacteroides* [58,59,60,61]. However, by definition, in HFDs, the carbohydrate amount in the total energy intake is decreased. Therefore, it is not clear whether microbiota composition and metabolism are mainly influenced by elevated fat or reduced carbohydrate content. In addition, more than their amount, the fat quality plays an important role in the gut microbiota composition. For example, HFDs are mainly composed of n-6 polyunsaturated fatty acids (PUFAs), often at the expense of n-3 PUFAs that have anti-inflammatory properties and modulate the intestinal microbiota in a beneficial way [62,63,64,65,66].

### 4.2. Proteins

Numerous studies have demonstrated that protein consumption positively correlates with overall microbial diversity [57,67]. For example, whey and glycated pea protein supplementation for a few days increases the commensal *Bifidobacterium* and *Lactobacillus* and decreases the pathogenic *B. fragilis* and *Clostridium perfringens* in the human gut [68,69]. Furthermore, pea protein consumption stimulates SCFA production in the intestine, leading to anti-inflammatory effects and mucosal barrier maintenance [70]. Conversely, a recent systematic review showed that short-term (1 to 4 weeks) beef intake has little or no effect on microbial profiles in humans [71]. However, when consumed at higher than recommended amounts, as part of a diet rich in sugar or fat, beef negatively affects the gut microbiota [71].

### 4.3. Carbohydrates

Modulation of the amount and/or type of carbohydrates in the diet for more than 4 weeks can alter the human gut microbiota and its metabolic products [72,73]. Carbohydrates can be subdivided into two categories: digestible and non-digestible. Digestible carbohydrates include starches and sugars (e.g., glucose, fructose, sucrose, and lactose) that are degraded in the small intestine by different enzymes. Upon digestion, they release glucose in the bloodstream and stimulate insulin production. Humans consume high amounts of glucose, fructose, and sucrose that increase the relative abundance of *Bifidobacteria* and reduce *Bacteroides* [74]. Non-digestible carbohydrates, such as fibers and resistant starch (RS), are not enzymatically degraded in the small intestine. Currently, four RS types (RS1-RS4) are recognized and may affect the bacterial composition differently [75]. Starch is a complex polysaccharide consisting of a mixture of amylose and amylopectin. The relative proportion of amylose and amylopectin affects the capacity of bacterial species to use different starch types for growth [76]. Dietary fibers, functionally known as microbiota-accessible carbohydrates, are present in inadequate amounts in Western diets [77]. As microbiota-accessible carbohydrates are the main source of energy for gut bacteria, their abundance and variety can influence gut microbiota composition and function [74]. Therefore, fibers should be considered prebiotics, which by definition are non-digestible dietary components that bring benefit to the host health via selective stimulation of the growth and/or activity of some microorganisms.

### 4.4. Prebiotics

In recent years, many researchers have tried to elucidate the relationship between prebiotics and human health [78]. Prebiotics can be metabolized by the intestinal microbiota. Moreover, their degradation products are SCFAs that are released in the blood, thus influencing not only the GI tract but also other distant organs [78]. Prebiotics include soybeans, inulins, unrefined wheat and barley, raw oats, and non-digestible oligosaccharides, such as fructans, polydextrose, fructo-oligosaccharides, galacto-oligosaccharides, xylo-oligosaccharides, and arabino-oligosaccharides. Fructo-oligosaccharides and galacto-oligosaccharides are two important groups of prebiotics with beneficial effects on human health [78]. A low fiber/prebiotic diet reduces the total bacterial abundance [79]. Moreover, prebiotics are routinely screened for their ability to selectively promote Bifidobacterial growth [74]. For more information on how prebiotics modulate the gut microbiota, the reader can refer to the recent review by Davani-Davari et al. [78].

### 4.5. Probiotics

Probiotics are live microorganisms that, when administered in adequate amounts, confer a health benefit to the host. Their effects on the gut and immune system are the most researched applications [80]. For example, fermented foods containing lactic acid bacteria, such as milk products and yogurt, represent a source of ingestible microorganisms that may beneficially regulate intestinal health and even treat or prevent IBDs [81]. *Lactobacillus*, *Bifidobacterium* and *Saccharomyces* strains have been safely and effectively used as probiotics for a long time. *Roseburia* spp., *Akkermansia* spp., *Propionibacterium* spp. And *Faecalibacterium* spp. are also promising probiotic microorganisms [82]. Research on the mechanisms of probiotic effects is mainly based on in vitro and animal models. However, the results cannot always be translated to humans, for instance, regarding probiotics for Crohn’s disease and mental health (see Jäger et al. and Sanders et al. for reviews) [80,82].

### 4.6. Bioactive Non-Nutrient Plant Compounds

Some bioactive non-nutrient compounds present in fruits, vegetables, grains, and other plants have been linked to a reduction in the risk of major chronic diseases [83]. These plant compounds include prebiotics and probiotics, as well as several chemical compounds, such as polyphenols (the largest group) and derivatives, carotenoids, and thiosulfates. Polyphenols can be subclassified into four main groups: flavonoids (including eight subgroups), phenolic acids (e.g., curcumin), stilbenoids (e.g., resveratrol), and lignans [84]. They promote health by limiting oxidative stress [85]. Common polyphenol-rich food types include fruits, seeds, vegetables, tea, cocoa products, and wine. The relative abundance of *Bacteroides* is increased in people consuming red wine polyphenols [86]. Moreover, it has been reported that the abundance of pathogenic *Clostridium* species (*C. perfringens* and *C. histolyticum*) is reduced after regular consumption of fruit, seed, wine, and tea polyphenols [74]. In a recent review, Rajha et al. showed that polyphenol metabolites interact with gut microbiota and mitochondria to fight many diseases, such as obesity, depression, inflammation and allergy [85].

### 4.7. Vitamins

Some vitamins are directly produced by the gut microbiota, and others play a role in modulating the presence of beneficial/detrimental bacterial species. Specifically, vitamin A can modulate health-beneficial microbes of the *Bifidobacterium*, *Lactobacillus* and *Akkermansia* genera [87,88]. Some B-complex vitamins are produced by gut commensals, and some of them contribute to increasing the virulence/colonization of potentially pathogenic microbes [88]. Vitamin C, D, and E supplementation may alter the microbiota composition by increasing the concentration of beneficial species, such as *Bifidobacterium* and *Lactobacillus*. Thus, vitamin intake could have a significant role in modulating gut microbiota. Moreover, this effect might depend on the host’s pre-supplementation vitamin level. However, clinical trials are still necessary to avoid adverse effects due to excess vitamin intake.

## 5. Potential Links between Gut Microbiome and Physical Fitness/Sports Performance

### 5.1. The Athletes’ Gut Microbiota, a Specialized Microbiota?

It is acknowledged that gut microbiota changes depend on individual factors, particularly in athletes, including energy expenditure, diet, drug intake (especially antibiotics) [89]. Many data on the gut microbiota composition in athletes are now available. As research in this field is rapidly expanding, the latest advances and major questions are summarized in several reviews [80,90,91,92,93,94,95] (Figure 2).

The analysis of literature data on this topic shows that the gut microbiota of athletes/exercised people is different from that of other populations and displays higher microbial diversity (Table 1). In 2014, Clarke et al. were the first to demonstrate that microbial diversity is increased in elite rugby players compared with matched controls. Specifically, the abundance of the phylum Bacteriodetes was decreased, whereas that of the genus *Akkermensia* was increased in athletes with low body mass index (BMI) (<25 kg/m^2^) compared with the high BMI (>28 kg/m^2^) group [96]. Moreover, Estaki et al. showed that peak oxygen uptake (VO_2_peak), the gold standard measure of cardiorespiratory fitness, can account for more than 20% of the variation in taxonomic richness in healthy men and women, after adjusting for all other factors, including diet [97]. Indeed, the abundance of key butyrate-producing taxa (Clostridiales, *Roseburia*, *Lachnospiraceae*, and *Erysipelotrichaceae*) was increased in individuals with high VO_2_peak values. The next step was to investigate whether the gut microbiota in athletes presents a specific composition in the function of the sports discipline. In 2019, Scheiman et al. generated much attention [98] by showing that the relative abundance of *Veillonella* was increased after a marathon and that the inoculation of a strain of *Veillonella atypica* from runner stool samples into mice significantly increased exhaustive treadmill run time in the inoculated animals. They also demonstrated a mechanistic link with lactate metabolism [98]. These results led to comparisons of the gut microbiota composition of athletes from different sports disciplines. The recent reviews by Mohr et al. in 2020 and by Aya et al. in 2021 concluded that in most cases, the gut microbiota α and β diversity are not different among sports disciplines, but some differences can be highlighted in the prevalence of some genera or taxa [90,95]. For example, O’Donovan et al., using shotgun metagenomic sequencing of fecal samples from elite athletes in 16 different sports, concluded that microbial diversity does not differ among sport disciplines [99]. However, they observed greater abundance of *Bifidobacterium animalis*, *Lactobacillus acidophilus*, *Prevotella intermedia* and *F. prausnitzii* in athletes with high dynamic components (high VO_2_max), and greater abundance of *Bacteroides caccae* in athletes with both high dynamic and static components (in relation with the maximal voluntary contraction component) [99].

Besides their chronic training regimes, the dietary intake patterns of athletes are often different from those of sedentary subjects [101], as is medication intake [102,103]. These factors also might influence their gut microbiota composition [94,104,105]. Finally, some data show that prolonged excessive exercise could have a detrimental effect on intestinal function [106]. Indeed, strenuous and prolonged exercise increases intestinal permeability and alters the gut barrier function. This promotes bacterial translocation from the colon [107,108], leak of bacterial LPS into the bloodstream, and activation of systemic inflammation. GI symptoms (e.g., abdominal pain, nausea, and diarrhea) are reported by 70% of athletes after strenuous exercise, and the frequency is higher in elite athletes than in recreational exercisers [109]. Besides the overall “diverse and rich” gut microbiota in athletes, many discrepancies can be observed in the microbiota profiles at lower taxonomic levels in relation to many confounding factors linked to the exercise type (e.g., intensity, mode, contraction type, duration, frequency), diet, drug intake, living environment, season, and many others. 

### 5.2. Gut Bacteria Are Sensitive to Physical Activity Modulation: Lessons from Training and Hypoactivity Models

Intervention studies using different training modalities in healthy sedentary participants and in populations with specific conditions (e.g., age-related pathologies, GI diseases, metabolic or inflammatory diseases, such as obesity or osteoarthritis) indicate that exercise and physical activity have beneficial effects on the gut microbiota [94,110,111]. Similarly, the many available studies in animal models (e.g., controlled diet) show that physical activity can modify the intestinal microbiota composition, particularly the bacterial richness and diversity [112,113,114,115,116] (Table 2). At lower taxonomic levels, the available findings highlight some discrepancies because gut microbiota responds differently to different training modalities: forced or spontaneous exercise [114,117,118], high-intensity interval training (HIIT), and moderate-intensity continuous training (MICT) [116,119]. Moreover, it seems that the changes observed in the fecal microbiota are more important in younger animals [115].

Not many longitudinal studies are available in humans. In their review of longitudinal studies on changes in specific groups of bacteria after initiation of an exercise or training program, Aya et al. found that BMI is a determining factor in the human microbiota response to exercise [90]. For instance, Allen et al. reported that after six weeks of supervised aerobic training, discrete incremental changes of the relative abundance of the Actinobacteria, Bacteroidetes, Firmicutes, Proteobacteria, and Verrucomicrobia phyla can be observed in the fecal microbiota of apparently healthy individuals with a BMI > 25 kg/m^2^. Conversely, gut microbiota from lean subjects responds to aerobic exercise by increasing the abundance of *Faecalibacterium* spp. and *Lachnospira* spp. and by reducing *Bacteroides* members [117]. To date, only one study reported findings related to HIIT. This non-randomized trial found that the abundance of the *Subdoligranulumwa* genus is increased in lean men after three weeks of cyclo-ergometer workout [131]. Thus, despite a growing literature, the effect of different training modalities on the human intestinal microbiota is still under investigation.

On the other hand, the effects of hypoactivity on the human gut microbiota have been rarely studied (Table 3). In their cross-sectional study, Bressa et al. compared healthy premenopausal active and sedentary women by correlating energy expenditure, physical activity intensity (measured with an accelerometer) and 16S gut microbiota sequencing data. They found that sedentary parameters were inversely correlated with the microbiota richness (i.e., number of species) and Shannon and Simpson indices [132].

Few studies have assessed the longitudinal impact of hypoactivity on the gut microbiota composition, and some concern space medicine. Data from microgravity studies suggest that gut bacteria are sensitive to drastic hypoactivity because spaceflight affects the microbial composition of the astronauts’ GI tract [134,135,136,137,139], mostly genera belonging to the Firmicutes phylum, the Clostridiales order, and the *Lachnospiraceae* family. Head-down bedrest and dry immersion are considered reliable ground-based models to study the physiological effects of hypoactivity in humans [140,141,142] and the response by gut microbiota to reduced/absence of physical activity. Nevertheless, very few studies have been published. Interestingly, *Lachnospiraceae* operational taxonomic units (OTUs) are increased in a mouse model of hypoactivity (hindlimb unloading) [133]. Similarly, Jollet et al. showed in healthy men that a short period of severe hypoactivity (five days of dry immersion that was enough to induce skeletal muscle atrophy) increases the OTUs associated with the Clostridiales order and the *Lachnospiraceae* family that belong to the Firmicutes phylum, without any effect on α and β diversity indices [138]. Moreover, propionate, an SCFA metabolized by skeletal muscle, was significantly reduced in the stool samples collected after the hypoactivity period. Overall, despite the limited number of available data, these first studies suggest that the *Lachnospiraceae* family is particularly sensitive to hypoactivity and might play a key role in the hypoactivity-gut microbiota axis.

### 5.3. The Gut-Muscle-Adipose Tissue Axis

Using bacteria-free (germ-free condition) or microbiota-depleted (dysbiosis) models, researchers revealed the existence of a “cross-talk” between gut microbiota and adipose tissue [143], and more recently, skeletal muscle [144], two key tissues for athletic performance. 

The most spectacular finding is that after the transfer of the microbiota of obese mice in germ-free mice (without gut microbiota), inoculated mice develop a fat phenotype in the absence of any change in their diet [145]. Moreover, people with obesity are characterized by dysbiosis compared with lean subjects [75]. According to the energy harvest theory, the gut microbiota of obese mice (the composition of which is influenced by a high-calorie diet) can extract more calories than the microbiota of lean mice [40]. In addition, obesity-induced dysbiosis decreases the expression of fasting inducing adipose factor (FIAF), thus increasing the activity of lipoprotein lipase (LPL) that facilitates the transport and thus the storage of fatty acids by peripheral tissues [146]. Several studies have confirmed the link between a high-fat diet, fat mass, and intestinal microbiota dysbiosis [147,148,149,150]. The Firmicutes/Bacteroidetes ratio is positively correlated with BMI and has been considered a sign of dysbiosis in people with obesity [151]; however, it is now questioned [152]. Bacterial metagenome sequencing in some individuals with obesity revealed that the number of bacterial genes is decreased, and consequently, the α diversity and β diversity are reduced compared with healthy subjects [153]. As said previously, the response of gut bacteria to exercise training is different in individuals with normal weight and with obesity. Moreover, the gut microbiota pattern in individuals with obesity is associated with various taxonomic signatures. This is probably due to many different lifestyle-associated factors (e.g., diet, physical activity, food additives and drugs) that affect the microbiota composition and/or diversity.

The influence of the intestinal microbiota on skeletal muscle, which is a metabolic organ, has been fully established only some years ago. The first data came from studies on the gut microbiota influence in obesity and cancer. Backhed et al. were the first to show that axenic mice fed a high-lipid diet are protected from obesity by two distinct mechanisms [145]. The first is an increase in the AMPK protein kinase (a metabolic sensor) associated with the over-activity of CPT1 (a mitochondrial membrane transporter of fatty acids, particularly in muscles). The second mechanism concerns the overexpression of FIAF that inhibits circulating LPL and stimulates PGC-1α, a transcription factor involved in oxidative metabolism and mitochondriogenesis in the gastrocnemius muscles [145]. Then, Delzenne and Cani’s group strongly suggested the existence of a possible axis between intestinal microbiota and skeletal muscle by investigating the composition of the intestinal microbiota of a mouse model of severe leukemia with cachexia [144]. Cachexia is a wasting syndrome observed in many chronic pathologies and is characterized by significant weight loss, particularly in muscles. Bindels et al. [130] found that the genus *Lactobacillus* spp., which has anti-inflammatory properties, is drastically decreased in cachectic mice. Oral targeted probiotic supplementation (*Lactobacillus reuteri* 100-23e and *Lactobacillus gasseri* 311476e) prevented muscle atrophy in cachectic mice by decreasing the expression of E3 ligases (responsible for the ubiquitination of proteins and then their degradation by proteasomes) as well as of atrogin-1, muscle ring finger-1 (MuRF1) and cathepsin-L (atrophy markers) in the tibialis anterior and gastrocnemius muscles. Supplementation also reduced lysosomal autophagy, involved in muscle breakdown, via protein 1A/1B-light chain 3 (LC3) downregulation [144]. These results were subsequently confirmed [154,155] and suggest that the gut microbiota could play an essential role in maintaining muscle mass in severe/chronic pathologies. The first review on a possible link between intestinal microbiota and muscle atrophy was published by Bindels and Delzenne and is currently considered the reference on this topic [156].

Besides the impact on muscle mass regulation, some studies have evaluated the functional link between gut microbiota and skeletal muscles. For instance, Yan and colleagues, inspired by the results obtained by transferring the flora of obese mice to axenic mice, investigated the impact on the muscle phenotype of a microbiota transfer from fat pigs to axenic mice [157]. Recipient axenic mice tended to mimic the donor phenotype. Muscle lipogenesis was significantly increased (higher concentration of intramuscular triglycerides, higher expression of LPL and of the membrane transporter of the fatty acid translocase (FAT/CD36), and slightly increased expression of the insulin mediator SREBP-1c (sterol regulatory element-binding protein 1). Moreover, the muscle structure and function of recipient axenic mice were closer to the donor muscle phenotype: slightly reduced cross-sectional area of muscle fibers with a shift towards the slow fiber type [157]. Nay et al. showed using ex vivo contractile tests that gut microbiota depletion (induced by antibiotic treatment) affects the intrinsic contractile muscle endurance associated with glucose homeostasis dysfunction [158]. These deleterious effects were normalized by natural microbiota reseeding.

On the basis of the available data, it is now evident that gut bacteria are essential for skeletal muscle and for the adaptation to exercise and training through numerous and various functions: control of the inflammatory and redox pathways, regulation of nutrient availability and metabolite-derived bacterial production, interaction with anabolic and catabolic processes, and also mitochondrial biogenesis, redox and immune system regulation [92,106,119,159,160,161].

Collectively, the latest studies on the athletes’ gut microbiota and the relationships between intestinal microbiota, physical activity and the gut-muscle-adipose tissue axis open innovative and original perspectives for rehabilitation and sports training in the context of individual performance optimization. The scientific community is now much interested in understanding the effects on the gut microbiota of supplementation with probiotics or, more generally, of “biotic diets” coupled or not with physical activity or training.

The take home messages, as well challenges and future directions dealing with independent impacts of diet or physical activity on the regulation of the gut microbiota are summarized in the Box 2.

Box 2Diet OR Physical Activity as Major Regulators of Gut Microbiota Composition.
✓The amount, type, and balance of the main dietary macronutrients, including n-3 PUFAs and non‐digestible carbohydrates, greatly influence the gut microbiota.✓Many dietary compounds are available to modify the gut microbiota composition. The intake of prebiotics and probiotics should be adapted to each patient’s characteristics.✓The diverse and rich gut microbiota in athletes must be better described at lower taxonomic levels to detect differences among sports disciplines.✓Gut bacteria are sensitive to sedentary behaviors (e.g., hypogravity), but very few data are available, and this issue needs to be thoroughly investigated. Some interventional studies using different training modalities to optimize gut microbiota composition in healthy sedentary people or in disabled populations show interesting results. Studies on the underlying mechanisms highlighted a cross‐talk between organs (i.e., gut-muscle-adipose tissue axis).


## 6. Gut Microbiota Modulation by Exercise and Nutrition for Health and/or Performance

As previously shown, gut microbiota diversity and function are affected by diet modulation and physical activity, and these changes influence the host’s physiology. Therefore, gut microbiota modulation appears as an appropriate target for nutritional and/or physical activity interventions to improve health and/or performance. Paradoxically, although both interventions are classically accepted and implemented, no human study has associated diet modulation with a physical activity program, with the exception of those combining probiotic consumption and high-level sports practice [80]. In their review, Donnati Zeppa et al. [162] mention the potential interest of different nutritional supplements for modulating gut microbiota composition in the hope of improving athletic performance; however, no human study has actually compared the combination of controlled physical training and nutritional intake on gut microbiota composition changes. Additional and synergistic effects could be expected, as suggested by recent studies in animals [113,116]. 

### 6.1. Diet and Microbiota Modulation, Health and Performance

Currently, many studies have tested specific diets (e.g., ketogenic diet, high-carbohydrate diet, high-protein diet, gluten-free diet) to improve sports performance; however, none has really investigated their potential influence on gut microbiota modulation in athletes, but only in active, non-sport populations or in populations with specific diseases, particularly chronic inflammatory diseases. For example, the very low carbohydrate ketogenic diet (VLCKD) is currently becoming very popular as a potential therapy for obesity and related metabolic disorders [163,164], and its effect on the gut microbiome is well documented [165,166,167,168]. In a sports context, VLCKD is used to favor fat oxidation during exercise, with the aim of delaying fatigue onset by sparing glycogen stores [169]; however, no data is available on its potential impact on the intestinal microbiota of athletes. Similarly, a high carbohydrate diet, which is often consumed by athletes 3–4 days before an endurance event and can sometimes represent more than 70% of the total energy intake, has not been studied as a potential gut microbiota modulator.

### 6.2. Probiotics, Athletes, and Performance

GI complaints are very common among endurance athletes and include nausea, vomiting, abdominal angina, and bloody diarrhea. Immune depression in athletes has also been described after excessive training load and has been associated with psychological stress, disturbed sleep, and extreme environmental conditions, all of which increase the risk of respiratory tract infections. In recent years, the use of probiotics in sports has been growing, mainly to attenuate GI symptoms and respiratory tract infections [80,170]. Overall, a positive effect has been demonstrated [170]. However, results are heterogeneous in the function of the probiotic strain, dose, intake duration, and even dosage form (capsules, sachets, or fermented milk). It seems that multi-strain probiotics in sachets or as fermented food consumed for a long period give better results (reduction of GI symptoms and respiratory tract infections) [170]. The probiotic effectiveness is probably explained by the increased barrier function, higher immune cell activity (from the pro-/anti-inflammatory pathways and immunoglobulin production), enhanced SCFA production, lower intestinal pH, and greater mucus production [171,172]. In athletes, probiotics may improve immune function by increasing interferon gamma production by T lymphocytes and possibly by increasing immunoglobulines A production by B lymphocytes [170]. Moreover, Lamprecht et al. suggested that probiotics favor Toll-like receptor 2 (TLR2) activation, which stimulates tight junction protein production, especially zonulin [173], thus decreasing gut permeability and consequently endotoxemia and GI symptoms.

Probiotics also influence gut microbiota composition. For example, West et al. found a 7-fold increase of the *Lactobacillus* genus after 11 weeks of *L. fermentum* supplementation in competitive cyclists (both sexes) [174]. Similarly, Martarelli et al. showed that in athletes, supplementation with *Lactobacillus* species during 4 weeks of intense physical activity significantly increased the fecal *Lactobacillus* count [175]. Unfortunately, these results were not associated with an increase in physical capacities. Currently, some data show that probiotics may have, directly or indirectly, positive effects on human sports performance (reviewed by the International Society of Sports Nutrition in 2019) [80]. For example, Huang et al. showed in a double-blind placebo-controlled trial that 6-week supplementation with *Lactobacillus plantarum* TWK10 at low (3 × 10^10^ CFU) and high doses (9 × 10^10^ CFU) prolongs the time to exhaustion during an 85% VO_2_ max exercise in a dose-dependent manner and decreases serum lactate levels during exercise and recovery [176]. Recently, 4-week supplementation using another *Lactobacillus plantarum* strain (PS128) has been associated with a decrease in the concentration of muscle damage and oxidative stress systemic markers after a half-marathon in recreational runners without changes in their exercise capacity [177]. Thus, ergogenic results are not always clear; however, probiotics might help to improve recovery by promoting muscle repair via increased protein synthesis [178,179].

### 6.3. Combining Supplements and Physical Activity Programs for Better Health by Modulating Gut Microbiota

Lifestyle interventions, including diet and/or physical activity programs, help to improve the metabolic profile of patients with obesity, type 2 diabetes, IBD, and other metabolic diseases. The American College of Sports Medicine recommends regular (150 min/week) low- to moderate-intensity continuous training for patients with obesity, (pre)diabetes, and other metabolic problems [180]. Currently, HIIT, defined as short bursts of intense activity interspersed by periods of low-intensity exercise or rest [181], is considered a time-efficient and safe exercise mode to reduce total fat mass, especially intra-abdominal fat mass [182,183]. It may also improve glucose metabolism [184,185,186]. Furthermore, HIIT modulates gut microbiota composition in humans and rodents [112,113,114,115,187]. A growing body of evidence suggests that in humans, dietary interventions, including supplementation of creatine monohydrate, caffeine, nitrate, sodium bicarbonate, beta-alanine, proteins, and essential amino acids, as well as manipulating carbohydrate availability, lead to more favorable outcomes after HITT programs by enhancing energy metabolism or by increasing the adaptive response during recovery [167]. However, none of them investigated the potential influence of the intervention on gut microbiota modulation. In this context, our group tested the impact on gut microbiota modulation of a 12-week HIIT program combined with a polyphenol-rich extract from five plants (olive leaves, bilberry, artichoke, chrysanthellum, and black pepper). This extract is called Totum-63 and was designed to reduce type 2 diabetes risk factors by modulating body composition and whole-body glucose homeostasis [113]. We found that HIIT, combined with Totum-63 supplementation, alters the body composition and glycemic profile in a rat model of pre-obesity, specifically by modulating the intestinal mucosa-associated microbiota. In our experimental conditions, the HIIT + Totum-63 combination significantly limited body weight gain, without any energy intake modulation, and improved glycemic control. Body weight variation was correlated with the α diversity of the colon mucosa microbiota, and this correlation was higher in the HIIT + Totum-63 group. Moreover, the relative abundance of *Anaeroplasmaceae*, *Christensenellaceae* and *Oscillospira* was higher in the HIIT + Totum-63 group. Thus, the combination of HIIT and Totum-63 supplementation could be proposed for the management of obesity and prediabetes and also of other chronic pathologies that involve gut microbiota dysbiosis.

More recently, we evaluated the effects of a 12-week intervention program combining physical activity (HIIT) and n-3 PUFA supplementation (i.e., the addition of linseed oil (LO) in the diet) on body composition and metabolic profile changes in a rodent model of obesity. We hypothesized that each intervention could specifically affect the mucosa-associated gut microbiota composition (α and β diversity) with more favorable adaptations in the HIIT + LO group [116]. It is known that n-3 PUFAs from natural sources or dietary supplements exert a beneficial effect on body composition and inflammation status [188,189], improve the intestinal barrier function [63] and integrity, and increase healthy bacterial communities [62,190]. By combining HIIT and LO, we hoped to induce an additive or even synergistic effect on the intestinal mucosa-associated microbiota that would promote body composition and metabolic profile changes. Our results show that HIIT significantly reduced total body fat mass and that the HIIT + LO combination improved alpha-linolenic acid to docosahexaenoic acid conversion and increased the relative abundance of *Oscillospira* bacteria in the colon microbiota [116]. *Oscillospira* abundance was negatively correlated with weight and fat mass gain. *Prevotella* also increased in the HIIT and HIIT + LO groups (compared with controls), and its abundance was negatively correlated with weight and fat mass gain. Thus, the combination of HIIT and LO could be proposed for the management of metabolic diseases, such as obesity.

The take home messages, as well challenges and future directions dealing with the Nutrition-Microbiota-Physical Activity triad is summarized in the Box 3.

Box 3The Nutrition‐Microbiota‐Physical Activity Triad.
✓Animal studies suggest additional and synergistic effects of physical activity and nutritional modulations on the gut microbiota composition that need to be confirmed in human studies. No study has investigated the effect of the high‐carbohydrate and very low ketogenic diets on the athletes’ microbiota.✓Multi-strain probiotics in the form of sachets or fermented food and consumed for a long period show beneficial effects in athletes.✓Lactobacillus plantarum species are the only probiotics with ergogenic effects in a double-blind-controlled human study. High-intensity interval training associated with n-3 PUFA or polyphenol-rich extract supplementation is the only tested training + diet intervention.✓The many possibilities offered by “biotic diets” and training modalities need to be investigated to show the clinical and/or ergogenic value of the triad.✓Triad-targeted interventions must take into account the microbiome profile of the patient or athlete to be efficient.


## 7. Conclusions and Perspectives

Today, there is no doubt that the discovery of the gut microbiota community opened a promising and rapidly growing research field on the potential beneficial health effects of manipulating the gut microbiota. Indeed, the gut microbiota influences the function of the intestine and also brain and metabolic tissues, such as adipose tissue and skeletal muscle. Through its bacteria-derived metabolites, the gut plays the role of an orchestra conductor in the host. Cumulative data in murine models allowed “failed/dysbiotic” and “healthy/competitive” microbiota profiles to be identified. Conversely, human studies are still limited. In many countries, National Gut Human Projects (e.g., the American Human Microbiome Project and the European Human Microbiome Action that started in 2021) have been set up to collect human fecal samples and to correlate the obtained microbiota results with the host’s characteristics. Human fecal samples and metagenomic data are currently collected for future biostatistics analyses. For instance, the Million Microbiome of Humans Project (MMHP) is a major international project, the aims of which are to create the largest human microbiota database in the world, to analyze 1 million samples, and to explore the full microbiome potential. One major milestone will be to launch interventional studies to modulate the gut microbiota composition because, as we demonstrated in this review, the gut can adapt its bacterial community in response to external factors, such as nutrition and physical activity. The scientific and medical communities must now find the best way(s) to optimize the nutrition-gut microbiota-physical activity triad for each patient or athlete. Currently, HIIT with n-3 PUFA or polyphenol-rich extract supplementation appears to be a promising combination. However, the possibilities offered by biotic nutrition and training modalities represent a veritable “playground” for scientists. The challenge is to develop innovative, original and promising microbiota-based strategies coupled with physical activity programs to optimize sports performance and medical treatments or to delay disease onset (Figure 3). Finally, it is essential to increase the population’s awareness of the need for a healthy diet and some physical activity for a healthy microbiota, although the triad mechanisms have not been fully elucidated yet. Some scientific organizations and large food companies are already campaigning about the importance of a healthy diet as a key factor in microbiota formation. However, they did not include the physical activity component. Indeed, the goal should be to make clear that both a healthy/well-balanced diet and regular (high-level) physical activity practice are needed to improve gut microbiota composition/function for better health and/or performance. We think that outreach programs should also include the triad concept to develop individualized microbiota-based strategies for health and sports performance management.

## Figures and Tables

**Figure 1 nutrients-14-00924-f001:**
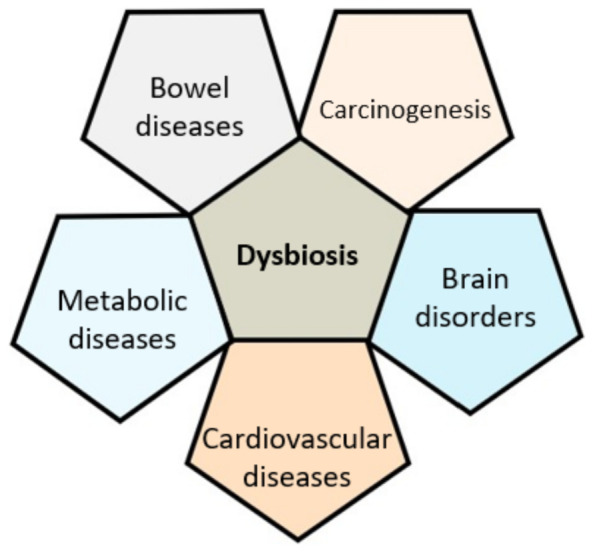
Disease-related gut microbiota dysbiosis.

**Figure 2 nutrients-14-00924-f002:**
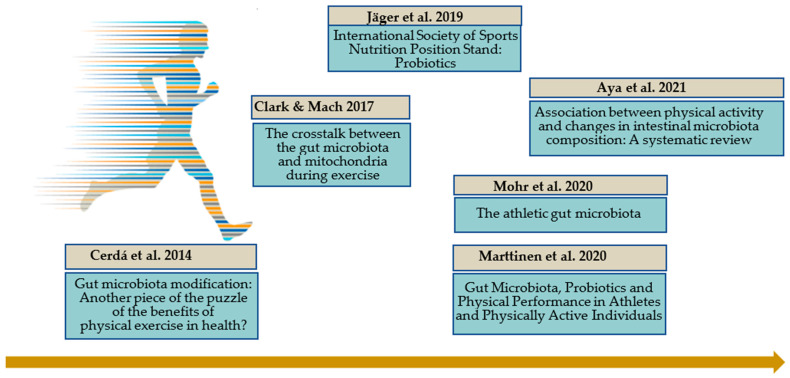
Athletic gut microbiota: main reviews [80,90,91,92,94,95].

**Figure 3 nutrients-14-00924-f003:**
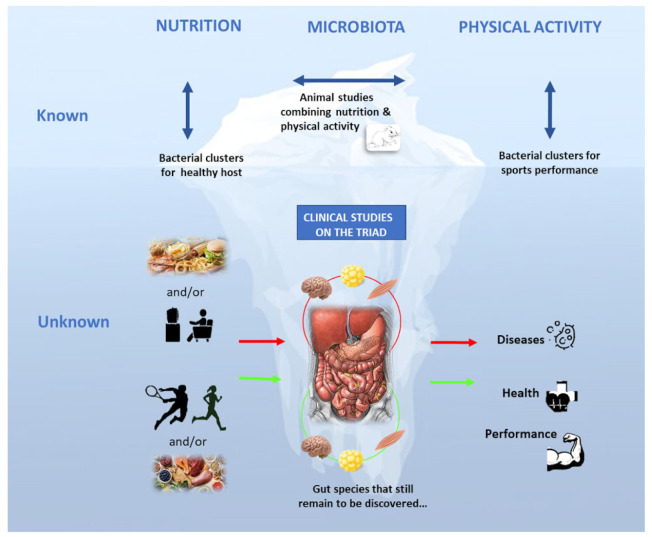
The “Nutrition-Physical activity-Microbiota” triad for health and sports performance: what is known and what remains to be discovered.

**Table 1 nutrients-14-00924-t001:** Gut microbiota in athletes—original articles.

Sports Type/Activity Level	Main Results	Authors-Year
Professional rugby players High level	Gut microbiota with higher richness, decrease of the phylum Bacteroidetes and increase of the genus *Akkermansia* prevalence.	Clarke et al., 2014 [96]
39 healthy participants Various cardiorespiratory fitness levels with similar age, body mass index, and diets	VO_2_ peak explained 20% of the variation in taxonomic richness. Increased abundances of key butyrate-producing taxa (Clostridiales, *Roseburia*, *Lachnospiraceae*, and *Erysipelotrichaceae*) in physically fit participants and increase in butyrate levels.	Estaki et al., 2016 [97]
Boston Marathon participants Treadmill-exercised C57BL/6 mice with acute supplementation of *Veillonella atypica*	Higher prevalence of *Veillonella atypica* in the athletes’ fecal samples after the marathon. Inoculation of this strain in mice significantly increased exhaustive treadmill run time via its metabolic conversion of exercise-induced lactate into propionate.	Scheiman et al., 2019 [98]
Bodybuilders and distance runners compared to healthy sedentary men	Gut microbiota α and β diversity similar in the two athlete groups. At the genus and species level, differences between sport disciplines, but associated with diet variations.	Jang et al., 2019 [100]
37 elite athletes who competed in 16 different sports/Olympic level	The gut microbiome and metabolome differ among sports, classified in groups. Diet is not the driver of these differences.	O’Donovan et al., 2020 [99]

**Table 2 nutrients-14-00924-t002:** Effect of training on gut microbiota—animal studies.

Training Modalities	Main Results on Gut Microbiota Composition	Authors-Year
Wistar male rats Standard diet ad libitum Spontaneous exercise: wheel, 5 weeks	↑ SM7/11 and T287 (Firmicutes)	Matsumoto et al., 2008 [118]
C57BL/6J male mice Standard diet ad libitum Spontaneous exercise: wheel, 5 weeks	↑ Lactobacillales and Bacillales (Firmicutes) ↓ Clostridiales (Firmicutes), Bacteroidales (Bacteroidetes), and *Erysipelotrichales* (Tenericutes)	Choi et al., 2013 [120]
Sprague-Dawley male rats Standard diet ad libitum Spontaneous exercise: wheel, 6 days	↑ *Lactobacillus*, *Bifidobacterium* and *Blautia coccoides*-*Eubacterium rectale* group	Queipo-Ortuño et al., 2013 [121]
8-weel-old C57BL/6J male mice Standard diet or high-fat diet Wheel 7 m.min^−1^ 60 min, 5 sessions/week for 14 weeks	↑ Firmicutes, *Lachnospiraceae*, *Peptostreptococcaceae*, *Pseudomonadaceae*, *Cryomorphaceae*, *Phyllobacteriaceae*, *Alcaligenaceae*, *Rhizobiaceae,* *Incertae_Sedis*_IV, *Microbacteriaceae*, *Nocardiaceae*, *Coriobacteriaceae*, *Flavobacteriaceae*, *Sphingobacteriaceae*, *Bradyrhizobiaceae*, *Burkholderiaceae*, *Comamonadaceae* ↓ Bacteroidetes, *Streptococcus* (HFD), Tenericutes (standard diet), *Porphyromonadaceae*, *Peptococcaceae*, *Streptococcaceae*	Kang et al., 2014 [122]
C57BL/6J male mice Low- or high-fat diet Spontaneous exercise: wheel, 12 weeks	↑ Bacteroidetes/Firmicutes ratio, *Clostridiaceae*, *Lachnospiraceae*, *Ruminococcaceae*, S24-7 ↓ Actinobacteria, *Lactobacillaceae*, *Turicibacteraceae*, *Erysipelotrichaceae*, *Bifidobacteriaceae*	Evans et al., 2014 [114]
24- or 70-day-old Fischer F344 male rats Standard diet ad libitum Spontaneous exercise: wheel, 6 weeks	Juvenile rats: ↑ Bacteroidetes, *Blautia* spp., *Anaerostipes* spp., *Methanosphaera* spp. ↓ Firmicutes, *Desulfovibrio* spp. and *Rikenellaceae* Adult rats: ↑ *Turicibacter* spp. and *Rikenellaceae*	Mika et al., 2015 [115]
6-week-old C57BL/6J male mice Standard diet Spontaneous exercise: Wheel, 30 days Controlled treadmill exercise: 8–12 m.min^−1^, 5% slope, 40 min, 5 sessions/week for 6 weeks	Spontaneous exercise: ↑ *Anaerotruncus* and ↓ *Prevotella* Controlled exercise: ↑ Tenericutes, Proteobacteria, *Nautilia*, *Oscillospira* and *Dorea*	Allen et al., 2015 [123]
C57BL/6J male mice Treadmill HIIT, 3 sessions/week for 6 weeks after 6 weeks of high-fat diet	↑ Bacteroidetes/Firmicutes ratio, Bacteroidales, *Dorea* and ↓ *Clostridiaceae* in cecal samples ↑ Actinobacteria in duodenum and jejunum samples ↑ *Lactobacillus* in ileum samples ↑ Bacteroidetes, Bacteroidales, *Dorea* and ↓ *Clostridium* and *Lachnospiraceae* in colon samples ↑ Bacteroidetes/Firmicutes ratio in fecal samples	Denou et al., 2016 [124]
Wistar Male Rats Standard diet Treadmill MICT, 5 times/week for 12 weeks Treadmill HIIT, 5 times/week for 12 weeks	MICT training: ↑ *Parasutterella excrementihominis*, *Lactobacillus johnsonii*, *Bifidobacteriaceae*, *Erysipelotrichaceae, Clostridium geopurificans* HIIT training: ↑ *Clostridium saccharolyticum*, *C. geopurificans*	Batacan et al., 2017 [112]
C57BL/6J male mice Standard diet ad libitum Spontaneous exercise: wheel, 6 weeks	↑ *Anaerostipes, Akkermansia* spp., *Lachnospiraceae,* *Ruminococcus* spp., *Parabacteroides* spp. ↓ *Prevotella*	Allen et al., 2018 [125]
C57BL/6J male mice Standard diet ad libitum Controlled training: treadmill, 4 weeks	More bacterial diversity in the exercise group ↑ Butyricimonas, *Akkermansia*	Liu et al., 2017 [83]
C57BL/6J male mice Standard diet or high-fat diet Spontaneous exercise: wheel, 14 weeks	↓ Firmicutes/Bacteroidetes ratio	McCabe et al., 2018 [126]
CEABAC10 male mice High-fat diet Spontaneous exercise: wheel, 12 weeks	↑ *Anaerotruncus*, *Parabacteroides*, Unclassified *Desulfovibrionaceae*, *Oscillospira*, *Ruminococcus*	Maillard et al., 2019 [127]
C57BL/6J male mice Standard diet or high-fat diet Controlled training: treadmill, 2 months	↑ *Vagococcus* in training group with standard diet ↑ *Vagococcus* and ↓ *Proteus* in training group with high-fat diet	Ribeiro et al., 2019 [128]
C57BL/6J male mice Standard diet or high-fat diet Spontaneous exercise: wheel, 10 weeks	↑ Bacteroidetes and ↓ *Lactobacillus* in training group with standard diet; ↑ α diversity and ↓ *Lactobacillus* in training group with high-fat diet	Aoki et al., 2020 [129]
ICR male mice Standard diet HIIT running, 7 weeks	↑ TM7, *Dorea*, *Dehalobacterium* ↓ Proteobacteria, *Candidatus arthromitus*	Wang et al., 2020 [130]
Wistar male rats Standard diet or high-fat diet ± Totum-63 HIIT running, 12 weeks	↑ *Anaeroplasma*, *Christensenellaceae* in HIIT group ↑ *Anaeroplasma*, *Christensenellaceae*, *Oscillospira* in HIIT+Totum-63 group	Dupuit et al., 2021 [113]
Wistar male rats Standart diet or high-fat diet ± linseed oil HIIT running, 12 weeks	↑ *Prevotella, YS2, Anaeroplasma* and ↓ *Clostridiales* in HIIT group ↑ *Prevotella, YS2, Anaeroplasma, Oscillospira* and ↓ *Clostridiales* in HIIT + linseed oil group	Plissonneau et al., 2021 [116]

↑: increase; ↓: decrease; HIIT: High-Intensity Interval Training; MICT: Moderate-Intensity Continuous Training.

**Table 3 nutrients-14-00924-t003:** Effect of hypoactivity on gut microbiota.

Population/Hypoactivity Model	Main Results	Authors-Year
Premenopausal women (*n* = 40): 19 active and 21 sedentary	Inverse association between sedentary parameters and microbiota richness ↓ *Bifidobacterium* spp., *Paraprevotella*, *Roseburia hominis*, *Akkermansia muciniphila,* and *Faecalibacterium prausnitzii* in sedentary women	Bressa et al., 2017 [132]
Mice Hindlimb unloading for 28 days	Increased microbial evenness, but not richness in hindlimb unloading vs. control group? ↓ Bacteriodetes, ↑ Firmicutes At the class/order level, ↑ *Clostridia/*Clostridiales and ↓ *Bacteroidia/Bacteroidales* At the family level, ↑ abundance of *Lachnospiraceae* and ↓ abundance of *S24**–**7*	Shi et al., 2017 [133]
Microbial content of human samples collected pre- and post-flight evaluated on culturable bacteria (not the genomic profile)	↓ *Lactobacilli and Bifidobacteria* post-flight ↑ *Enterobacteria* and *Clostridia* post-flight	Crucian et al., 2018 [134]
The NASA Twins Study: twins (one on ground and the other in the International Space Station, for 25 months)	No impact on microbiome diversity Significant and spaceflight-specific increase in the Firmicutes/Bacteriodetes ratio	Garrett-Bakelman et al., 2019 [135]
Astronauts 6 to 12 months in the International Space Station	↑ Shannon α diversity and richness. Changes in 17 gastrointestinal genus abundance during spaceflight 13/17 genera belonged to the phylum Firmicutes, mostly to the order Clostridiales ↓ *Akkermansia*, *Ruminococcus*, *Pseudobutyrivibrio* and *Fusicatenibacter*	Voorhies et al., 2019 [136]
C57BL/6 female mice 37 days in the International Space Station	Unchanged richness of microbial community Higher Firmicutes/Bacteriodetes ratio with ↓ phylum Bacteriodetes), ↑ genera of the *Lachnospiraceae* family and *Ruminococcaceae* UCG-010 genus ↓ *Hydrogenoanaerobacterium* genus and *Tyzzerella* genus	Jiang et al., 2019 [137]
Healthy men (*n* = 14) Dry immersion for 5 days	Unchanged α and β diversity indices ↑ Clostridiales order and *Lachnospiraceae* family ↓ Propionate levels in post-dry immersion stool samples	Jollet et al., 2021 [138]

↑: increase; ↓: decrease.

## Data Availability

Not applicable.

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
