# Peer review of "The Nutrition-Microbiota-Physical Activity Triad: An Inspiring New Concept for Health and Sports Performance"

_nutrients, 2022, doi:10.3390/nu14050924_

Round 1
Reviewer 1 Report
The paper The triad nutrition-microbiota-physical activity: an inspiring new concept for health and sports performance, bring us a new google to watch the exercise influence on intestinal microbiota and function as well. This "new observation angle" will bring us new questions and investigations on this field.
Author Response
We thank the reviewer for his/her interest in this topic. We agree that this subject probably will grow in the near future.

Reviewer 2 Report
The topic is very relevant and I believe the manuscript is comprehensive supporting the recommendation for future publication. The manuscript may benefit focus on the hypothesis and less basic education of the reader (e.g., defining prebiotics, probiotics, phytonutrients, et cetera). Other publications could be cited and help concentrate the paper.
There is a significant amount of reporting of prior study findings. The authors would bring greater value to the audience explaining why that finding was important. I value the author's interpretation more than repeating the findings of others.
Although well written, please consider including a native English speaking editor to help clarify some language issues throughout the paper.

Author Response
The topic is very relevant and I believe the manuscript is comprehensive supporting the recommendation for future publication.
Response
We thank the reviewer for his/her interest in our work.
The manuscript may benefit focus on the hypothesis and less basic education of the reader (e.g., defining prebiotics, probiotics, phytonutrients, et cetera). Other publications could be cited and help concentrate the paper. There is a significant amount of reporting of prior study findings. The authors would bring greater value to the audience explaining why that finding was important. I value the author's interpretation more than repeating the findings of others.
Response
It seemed important to us to start our review by summarizing the knowledge on the mechanisms underlying the gut microbiota implication in various chronic pathologies, and also in sports performance (which is a more recent topic). In the second part, we focused on the combination of physical activity and nutritional modulations of gut microbiota composition in the context of health or performance, the hypothesis being that the interaction of the two mechanisms could potentiate their effects. This concept is quite new in the literature (hence the notion of a triad) and very few studies have combined the two interventions to assess their potential beneficial effects on health and/or performance. To clarify this hypothesis, the Introduction has been modified.
Furthermore, we included three “Take-home messages” boxes to clarify and highlight the main findings, challenges and future directions according to our interpretation as authors of the review.
Although well written, please consider including a native English speaking editor to help clarify some language issues throughout the paper.
Response
The review had already been proofread by a native English speaker, but a second revision was made to meet expectations.

Reviewer 3 Report
Dear Editor,
The authors wrote an excellent review of the state of the art on all the correlations related to the intestinal microbiota and nutrition within the physical activity panel. The article is valid because it offers a clear window on how nutrition and sporting activity affect the intestinal microbial population, and on the other hand how the latter is so prevalent for health. My opinion is to accept the job after minor revision. In particular, I would advise the authors to implement the introduction or add it to the second chapter which is always introductory in its own way. I also recommend implementing chapter 4 relating to diet, perhaps you could explore the various topics a little more, in particular probiotics and prebiotics that are so important for the microbiota. Also I recommend changing the title of paragraph 4.6 "other phytochemical" since they were not mentioned before. Also in this case, the authors could better talk about the countless molecules (vitamins above all) important for the health of humans and their microbiota. The conclusions should be deepened: how could this information be managed, in the sense that it would be appropriate to sensitize the population to a healthy diet and some physical activity in order to implement the health of the microbiota and therefore of humans? How could this information benefit chronic disease medical therapies? Perhaps there are already national programs in some countries that explain the correlations described in the work and could be described in the conclusions as a concrete application of the triad.
Author Response
The authors wrote an excellent review of the state of the art on all the correlations related to the intestinal microbiota and nutrition within the physical activity panel. The article is valid because it offers a clear window on how nutrition and sporting activity affect the intestinal microbial population, and on the other hand how the latter is so prevalent for health. My opinion is to accept the job after minor revision.
Response
We thank the reviewer for his/her interest in our work.
In particular, I would advise the authors to implement the introduction or add it to the second chapter which is always introductory in its own way.
Response
In line with the remarks of reviewers 2 and 3, the Introduction has been modified to better introduce the hypothesis of the cross-talk between physical activity/nutrition and gut microbiota.
I also recommend implementing chapter 4 relating to diet, perhaps you could explore the various topics a little more, in particular probiotics and prebiotics that are so important for the microbiota.
Response
To help/guide the interested reader, we added some major references of reviews on the influence of pre- and probiotics on gut microbiota composition and function.
Also I recommend changing the title of paragraph 4.6 "other phytochemical" since they were not mentioned before.
Response
The title has been modified and replaced by “Bioactive non-nutrient plant compounds”.
Also in this case, the authors could better talk about the countless molecules (vitamins above all) important for the health of humans and their microbiota.
Response
A paragraph concerning vitamins has been added in chapter 4, as suggested.
The conclusions should be deepened: how could this information be managed, in the sense that it would be appropriate to sensitize the population to a healthy diet and some physical activity in order to implement the health of the microbiota and therefore of humans? How could this information benefit chronic disease medical therapies? Perhaps there are already national programs in some countries that explain the correlations described in the work and could be described in the conclusions as a concrete application of the triad.
Response
The conclusion has been rewritten according to the reviewer’s remarks, and recommendations were included. The goal is to sensitize these population to a healthy/well-balanced diet concomitantly with a regular (high-level) physical activity practice to improve gut microbiota composition/function for better health and/or performance.
